# Using street view imagery for 3D survey of rock slope failures

J., Voumard[1], A., Abellan[1,2], P., Nicolet[1,3], M.-A. Chanut[4], M.-H., Derron[1], M., Jaboyedoff[1]

[1] Risk analysis group, Institute of Earth Sciences, FGSE, University of Lausanne, Switzerland
[2] Scott Polar Research Institute, Department of Geography, University of Cambridge, United Kingdom
[3] Geohazard and Earth Observation team, Geological Survey of Norway (NGU), Norway
[4] Groupe Risque Rocheux et Mouvements de Sols (RRMS), Cerema Centre-Est, France

## Abstract

We discuss here different challenges and limitations on surveying rock slope failures using 3D reconstruction from image sets acquired from Street View Imagery (SVI). We show how rock slope surveying can be performed using two or more image sets using online imagery with photographs from the same site but acquired at different instants. Three sites in the French alps were selected as pilot study areas: (1) a cliff beside a road where a protective wall collapsed consisting of two images sets (60 and 50 images in each set) captured within a six years time-frame; (2) a large-scale active landslide located on a slope at 250 m from the road, using seven images sets (50 to 80 images per set) from 5 different time periods with three images sets for one period; (3) a cliff over a tunnel which has collapsed, using two image sets captured in a four years time-frame. The analysis include the use of different Structure for Motion (SfM) programs  and the comparison between the so-extracted photogrammetric point clouds and a LiDAR derived mesh that was used as a ground truth. Results show that both landslide deformation and estimation of fallen volumes were clearly identified in the different point clouds. Results are site and software-dependent, as a function of the image set and number of images, with model accuracies ranging between 0.2 and 3.8 m in the best and worst scenario, respectively. Although some limitations derived from the generation of 3D models from SVI were observed, this approach allow obtaining preliminary 3D models of an area without on-field images, allowing extracting the pre-failure topography that would not be available otherwise.

## Keywords

Street View Imagery (SVI), Structure from Motion (SfM), photogrammetry, 3D point cloud, natural hazard, landslide, rockfall.

## 1 Introduction

3D remote sensing techniques are becoming widely used for geohazard investigations due to their ability to represent the geometry of natural hazards (mass movements, lava flows, debris flows, etc.) and its evolution over time by comparing 3D point clouds acquired at different time steps. For example, 3D remote sensing techniques are helping to better quantify key aspects of rock slope evolution, including the accurate quantification of rockfall rates and the deformation of rock slopes before failure using both LiDAR (Rosser et al., 2005; Oppikofer et al, 2009; Royan et al., 2013; Kromer et al., 2015; Fey and Wichmann., 2016) and photogrammetrically derived point clouds (Walstra et al., 2007; Lucieer et al., 2013, Stumpf et al., 2015; Fernandes et al., 2016; Guerin et al., 2017; Ruggles et al., 2016).

Airborne and terrestrial laser scanner (ALS and TLS, respectively) are commonly used techniques to obtain 3D digital terrain models (Abellan et al., 2014). Despite their very high accuracy and resolution, these technologies are costly and often demanding from a logistic point of view. Alternatively, Structure from Motion (SfM) photogrammetry combined with multiview-stereo (MVS) allow using end-user digital cameras to generate 3D point clouds with a decimetre level accuracy in a cost-effective way in order (Westoby et al., 2012; Carrivick et al., 2016).

Whereas most of the studies in SfM literature utilise pictures that were captured on purpose (Eltner et al., 2016), the potential of using internet-retrieved pictures for 3D reconstruction has not been fully discussed before (e.g. Snavely et al., 2008; Guerin et al., 2017). One of the large sources of pictures on-line is the Street View Imagery (SVI) services, which offer 360 degrees panoramas from many roads, streets and other places around the world (Anguelov et al 2013). It allows to remotely observe areas without physically accessing them and so in a cost-effective way, with applications in navigation, tourism, building texturing, image localization, point clouds georegistration and motion-from-structure-from-motion (Zamir et al. 2010; Anguelov et al, 2010; Klingner et al, 2013; Wang, 2013; Lichtenauer et al., 2015).

The aim of present work is to ascertain up to which extent 3D models derived from SVI can be used to detect geomorphic changes on rock slopes.

## 1.1 Street View Imagery

The most common SVI service is the well-known Google Street View (GSV) (Google Street View, 2017) that is available from Google Maps (Google Maps, 2017) or Google Earth Pro (Google Earth Pro, 2013). We used both GSV as SVI service in this study. Alternatives include StreetSide by Microsoft (StreetSide, 2017) and other national services like Tencent Maps in China (Tencent Maps, 2017). SVI was firstly deployed in urban areas to offer a virtual navigation into the streets. More recently, non-urban zones can also be accessed, and were used for the analysis of rock slope failures in this manuscript.

GSV was firstly used in May 2007 for capturing pictures in streets of the main cities in USA and it has been deployed worldwide over the forthcoming years, including also rural areas. GSV images are collected with a panoramic camera system mounted on different types of vehicles (e.g. a car, train, bike, snowmobile, etc.) or carried into a backpack (Anguelov et al, 2010).

The GSV first generation camera system was composed of eight wide-angle lenses and it is currently composed of fifteen CMOS sensors 5Mpx each (Anguelov et al, 2010). The fifteen raw images, which are not publicly available, are processed by Google to make a panorama view containing an a priori unknown image deformation (Figure 1). A GSV panorama is normally taken at an interval of around ten meters along a linear infrastructure (road, train or path).

GSV proposes a *back-in-time function* on a certain number of locations since April 2014. In addition, other historical GSV images are available from 2007 for selected areas only. The number of available image sets greatly varies at different locations: while some places have several sets, many other locations have only one image set. Back in time function is especially useful for natural hazards because it is possible to compare pre- and post-events images.

The GSV process can be explained in four steps (Anguelov et al, 2010; Google Street View, 2017): 1) Pictures acquisition in the field; 2) Image alignment: preliminary coordinates are given for each picture, extracted from sensors on the Google car that measure GNNS coordinates, speed and azimuth of the car, helping to precisely reconstruct the vehicle path. Pictures can also be tilted and realigned as needed; 3) Creation of 360° panoramas by stitching overlapping pictures. Google applies a series of processing algorithms to each picture to attenuate delimitations between each picture and to obtain smooth pictures transitions; 4) Panoramas draping on 3D models: the three LiDAR mounted on the Google car

help to build 3D models of the scenes. 360° panoramas are draped on those 3D models to give
a panorama view close to the reality. Each picture of the panorama has its own internal
deformation, and the application of the processing chain described above makes inconstant
deformation in the 360° panorama; in addition, the end-user does not have any information or
control on it.

## 1.2 SfM-MVS

Structure for Motion (SfM) with Multi-View Stereo (MVS) dense reconstruction is a cost-
effective photogrammetric method to obtain a 3D point cloud of terrain using a series of
overlapping images (Luhmann et al., 2014). The prerequisites are that: (1) the studied object
is photographed from different points of view, and (2) each element of the object must be
captured from a minimum of two pictures assuming that the lens deformation parameters are
known in advance (Snavely 2008; Lucieer et al. 2013). If these parameters are not known
beforehand, three pictures is the minimum requirement (Westoby 2012), and about six
pictures is preferred. The particularity of SfM-MVS is that prior knowledge of both intrinsic
camera parameters (principal point, principal distance and lens distortion) and extrinsic
camera parameters (orientation and position of the camera centre (Luhmann et al., 2014)) is
not needed.
The workflow of SfM-MVS normally includes the following steps: 1) Feature detection and
matching (Lowe, 1999); 2) Bundle adjustment (Snavely et al., 2006; Favalli et al., 2011;
Turner et al., 2012; Lucieer et al., 2013); 3) Dense 3D point cloud generation (Furukawa et
al., 2010; Furukawa & Ponce, 2010; James & Robson, 2012); and 4) Surface reconstruction
and visualization (James & Robson, 2012).

## 2   Study areas and available data

We selected three study areas in France to generate point clouds from GSV images. This
country was chosen because GSV cover the majority of the roads and because the timeline
function works in most of the areas covered by GSV, meaning that several periods of
acquisition are available. Moreover, landslide events occur regularly on French alpine roads.
The aerial view of the three areas is shown in Figure 2A and examples of corresponding GSV
images in Figure 2B and 2C.
The first case study ("Basse corniche" site) is a 20 m high cliff beside a main road in
Roquebrune – Cap Martin connecting the town of Menton to the Principality of Monaco, in
South-Eastern France. A wall built to consolidate the cliff collapsed after an extreme rainfall
event in January 2014, blocking the road (Nice-Matin, 2014). Two 3D models were built with
60 GSV images taken in 2008 before the wall collapse, and 50 GSV images taken in 2014
after the event.
The second case studies is Séchilienne landslide, located 15 km South East of Grenoble (Isère
department, France). The active area is threatening the departmental road RD 1091
connecting the towns of Grenoble and Briançon as well as a set of ski resorts such as L'Alpe
d'Huez and Les Deux Alpes to the plain. This landslide is about 800 m long by 500 m high
and it has been active during more than thirty years (Le Roux et al. 2009; Durville et al. 2011;
Dubois et al. 2014). The shortest distance between the landslide foot and the former road was
250 m and the longest distance between the landslide head and the road is 1 km. A new road,
located higher in the opposite slope, has been opened since July 2016. Different SfM-MVS
processing were tested using from 50 up to 80 GSV images, at six different times from April
2010 to June 2015.
The third case study is located in "Arly gorges", between Ugine and Megève on the path
Alberville – Chamonix-Mont-Blanc. A rockfall of about 8'000 m$^3$ affected the road at the
entry of a tunnel on January 2014 (France 3, 2014). Different sets of images ranging from 60
to 110 GSV images were processed in order to obtain three 3D models of the road, the tunnel
entry and the cliff above the tunnel.
We used two image sets from for the first study site, eight image sets for the second study site
and four image sets for the third study site, with dates ranging from May 2008 up to
December 2016, as described in Table 1.

# 3   Methodology

First step to make SfM-MVS with SVI is to obtain images from a SVI service. GSV has been
used in this study (Figure 1). Given that original images of the Google cameras are not
available, one of the two ways to get images from GSV is to manually extract them from the
GSV panoramas. We took print screens (1920 x 1200 pixels, 2.3 Mpx) of GSV panoramas of
the studied areas at each acquisition step, separated by about ten meters, from Google Maps.
Several images were taken from the same point of view with different pan and tilt angles
(Figure 1C) when the studied object was too close to the road. In such cases, it was impossible
to have the entire area in one image because the image is not wide enough to capture the
entire studied area (for example a 10 m high cliff along road). When the studied area was far
away from the road, we took print screens of zoomed sections of the panorama.
To perform temporal comparisons on each site, images were taken at the different dates
proposed by GSV with pre- and post-event images sets. We used the SfM-MVS program
VisualSFM (Wu 2011) for dense point cloud reconstruction for the print screens images from
Google Maps and we used CloudCompare (Girardeau-Montaut 2011) for point cloud
visualization and comparison. Comparison between two point clouds was made using point-
to-mesh strategy. To this end, a mesh was generated from the reference point cloud (the point
cloud with the oldest images for site 1 or the LiDAR scans for sites 2 and 3) and then the
other point cloud was compared to this reference mesh. The computed shortest distance, a
signed value, between the mesh and the point cloud is the length of the 3D vector from the
mesh triangle to the 3D point. Thus, average distances and standard deviations for each
comparison of point clouds have been computed. Point density of point clouds was obtained
using the "point density" function in CloudCompare with the "surface density" option.
Beside the images taken from print screens as described above, we also obtained GSV images
(4800 x 3500 pixels, 16.8 Mpx) from Google Earth Pro on sites 2 and 3 with the "save image"
function. This second way to get GSV allows to get images with a higher resolution than print
screen images. Unfortunately, there is no timeline (or "back in time") function in Google
Earth Pro; it is only possible to save images from the last picture acquisition, i.e. generally
post-event images. GSV images from Google Earth Pro were processed with the Agisoft
PhotoScan software (Agisoft 2015) for dense point cloud reconstruction, which provides
much better results than VisualSFM. GSV images from Google Map were processed with
VisualSFM because Agisoft was not able to process those print screens. The flowchart of
Figure 3 shows the processing applied to both types of images (print screens and saved
images).
A rough scaling and georeferencing of the 3D point clouds was made without ground control
points, only with coordinates of few points extracted from Google Maps or from the French
geoportal (Géoportail, 2016).
It is important to mention here that a series of issues are expected when attempting to use SVI
for 3D model reconstruction with SfM-MVS. Indeed, GSV images are constructed as 360°
panoramas from a series of pictures, so the internal deformation of the original image is not
fully retained on the panoramas. In other words, the deformation of a cropped section of the
panorama will be a main function not only of the internal deformation of the camera and lens
but to the panorama reconstruction process; this circumstance will significantly influence the
bundle adjustment process and so to the 3D reconstruction.
In addition, GoPro Hero4+ images from a moving vehicle on the road were taken by the
authors on site 2, as well a series of images captured using a GoPro Hero5 Black camera
standing on site 3 (image resolution of 4000 x 3000 pixels, 12 Mpx). Six LiDAR scans were
also taken on site 3. This information was used for quality assessment purposes.

## 4   Results and discussion

Different results are obtained depending on the software used for SfM-MVS processing. For
all case studies, VisualSFM gave results with print screens from GSV in Google Maps while
Agisoft PhotoScan could not align those print screens despite adding a series of control points
measured with Google Earth Pro. Resolution of print screens images seem to be insufficient to
be processed with Agisoft PhotoScan. However, with higher point density and empty areas,
Agisoft PhotosScan provided better results with images from Google Earth Pro than
VisualSFM.

### 4.1   Site 1 – "Basse corniche" site

It was possible on "Basse Corniche" site to estimate the fallen volume by scaling and
comparing the 2008 (Figure 4A) and 2010 (Figure 4B) point clouds. The 2008 point cloud is
composed of 150'000 points with an average density of 290 points per square meter and the
2014 point cloud is composed of 182'000 points with an average density of 640 points per
square meter (Table 1). VisualSFM could align the images and make 3D models before and
after the wall collapse. It was possible to roughly scale and georeference the scene with the
road width and few point coordinates measured on Google Earth Pro or on the French
geoportal. After aligning the two 3D point clouds, meshes were built to compute the collapsed
volume. The point-to-mesh alignment in CloudCompare of both point clouds was done on a
small stable part of the cliff (Figure 4C) with a standard deviation of the point-to-mesh
distance of about 10 cm  (Figure 9 and Table 2) and on the entire cliff beside the vegetation
with a standard deviation of about 25 cm (Figure 4E). In the collapsed area, the maximal
horizontal distance between the two datasets is about 3.9 m (red colour in Figure 4D). The
collapsed volume (including a possible empty space between the cliff and the wall before the
event) was estimated to be about 225 m$^3$ using the point cloud comparison. Based on Google
Street images, we manually estimated the dimensions of this volume (15 m long x 10 m high
x 1.5 m deep), getting a similar value.
The obtained point clouds on site 1 allow to detect object of few decimetres. This accuracy
was adequate to estimate the collapsed volume with an accuracy similar to the estimation
made by hand based on the GSV photos and distances measured on Google Earth Pro and the
French geoportal. This relatively high accuracy is due to the following factors: good image
quality, reduced distance between the cliff and camera locations, good lighting conditions,
absence of obstacles between the camera location and the area under investigation, no
vegetation and efficient repartition of point of view around the cliff (Figure 2 A).
## 4.2 Site 2 – Séchilienne Landslide
Eight point clouds of which seven of SfM-MVS process with GSV images were generated for
Séchillienne landslide at six different time steps (from April 2010 to June 2015). Three
different image sources were used: GSV print screens from Google Maps, GSV images saved
from Google Earth Pro and images from a GoPro HERO4+ camera from a moving vehicle
(Figure 5 and Table 1). Two different programs (VisualSFM and Agisoft PhotoScan) were
used for image treatment in function of the image sources (Figure 3 and Table 1). The number
of 3D points on the landslide area varies from 9'500 to 22'500 points for a processing with
VisualSFM with an average density of 0.25 to 0.85 points per square meter, while 236'000
3D points were generated when using Agisoft PhotoScan with an average density of 2 points
per square meter (Table 1). In comparison, 1'500'000 points were obtained on the same area
using terrestrial photogrammetry with a 24 Mpx reflex camera.
Results were aligned on a 50 cm resolution airborne LiDAR scan of the landslide acquired in
2010. Then, the street view SfM-MVS point clouds were aligned and compared with a mesh
from the LiDAR scan using the point-to-mesh strategy. The alignment between the LiDAR
point cloud and SfM-MVS point clouds derived from SVI is a key factor to define the quality
of the clouds comparison. This alignment on stable areas (manually selected) was not easy to
perform because of the low density of points on the SfM-MVS clouds derived from SVI. We
noted a huge difference in the number of points between the different SfM-MVS clouds
derived from SVI. This difference on the number of points shows the impacts of the image
quality. Images with a good quality (resolution, exposition, sharpness) will give point clouds
with a higher number of points as point clouds from low quality images.
Comparison results between SfM-MVS point clouds derived from SVI and airborne LiDAR
scan highlight surface changes in the Séchilienne landslide over the years (Figure 8 and Table
1). The 2010 point cloud (Figure 5 A2) compared with 2010 LiDAR scan does not show any

significant changes. Orange and red colours small dots are spread out on the entire landslide surface suggesting artefacts and not a real slope change. The 2010-2011 point clouds comparison (Figure 5 B2) shows few little red colour pattern (materiel accumulation) in the deposition and in the failure areas. The 2016 point cloud (Figure 5 C2) highlights material deposition in red colour, in the left part. This is confirmed with comparison of a 2013 terrestrial LiDAR. The blue colour pattern indicate a loss of material in the failure and the toe areas. The 2014 point cloud (Figure 5 D2) shows similar results than the 2013 point cloud with however a light increase of material in the deposition area and rock loss in the failure area. The 2010 to 2014 point clouds (Figure 5 A-D) were process with VisualSFM with GSV print screens in Google Maps (Table 1).

Three 2015 point clouds were processed: the first with VisualSFM and GSV print screens (Figure 5E), the second with VisualSFM with GSV images from Google Earth Pro (Figure 5F) and the third with Agisoft PhotoScan with images form Google Earth Pro again (Figure 5G). The results should be the same for the three point clouds but we noticed significant differences. The 2015 point cloud processed with VisualSFM and GSV images from Google Earth Pro (4800 x 3500 pixels), has a higher point density than the 2015 point cloud processed with GSV print screens (1920 x 1200 pixels). The 2015 point cloud with Agisoft PhotoScan and images from Google Earth Pro has a point density significantly higher (Table 1). The accumulation material (red colour in the left part) in the deposition area is clearly observable on the three 2015 point clouds, as the rock displacement-toppling below the failure area (red colour pattern in the failure area viewed as a material accumulation from the road). The loss of material (blue colour) is also well observable in the failure area and, to a lesser extent, in the right part of the deposition area. The last 2015 point cloud is very similar to the 2016 GoPro point cloud (Figure 5 H2) which confirms the results of SfM-MVS processing with GSV images.

Results of site 2 show that images with low resolution and with low lighting generated a lower number of points compared to the models generated with the last generation of GSV cameras, having higher resolution, more advanced sensors and pictures taken with favourable lighting conditions. The large distance between the road and the landslide considerably limits the final accuracy due to low image resolution, as discussed in Eltner et al., 2016; the closest distance between the road and the centre of the landslide is 500 m and the largest distance between the upper part of the landslide and the point of view is about 1'400 m. Furthermore, the vegetation on the landslide foot and along the road as well as a power line partially

obstruct the visibility of the study area. In addition, clouds are present on several images on
the top of the scarp, degrading the upper part of the 3D point cloud.
4.3   Site 3 – Arly Gorges
Four point clouds of which three of SfM-MVS process derived from GSV images were
generated on the "Arly gorges" site, at four different times (from March 2010 to December
2016). Three different images sources (GSV print screens from Google Maps, GSV images
exported from Google Earth Pro and our own images acquired from a GoPro HERO5 Black)
were used (Figure 6 and Table 1). Two different programs (VisualSFM and Agisoft
PhotoScan) were tested. In addition, a LiDAR point cloud resulting from an assembly of six
Optech Ilris scans has been used as ground truth (Figure 6E). The number of points varies
from 35'000 points to 3.2 million points with an average density of 40 to 2'200 points per
square meter (Table 1).
The 3D point cloud from the "GoPro Hero5 Black" images has been roughly georeferenced,
scaled and oriented thanks to the GNSS chip integrated in the camera and has been controlled
and refined with points coordinates extracted from Google Maps and the French geoportal.
The three point clouds processed from GSV images and the LiDAR scan have been roughly
aligned to this reference. Then the four SfM-MVS point clouds (three with GSV images and
one with GoPro images) were precisely aligned and scaled on the LiDAR point cloud, which
was considered as the reference cloud.
The analysis (Figure 9, Tables 1 and 2) shows that the 2010 model derived from GSV images
processed with VisualSFM gives the least accurate results (Figures 6A and 7A): we hardly
perceive on that figure the wall of the tunnel entry and the wide cliff structures. The results of
the 2014 point cloud from GSV images processed with the same program are slightly better
(Figure 6B and 7B): the right-hand tunnel entry is modelled while it was not the case on the
2010 point cloud. The point cloud processed in Agisoft PhotoScan derived from 2016 GSV
images saved from Google Earth Pro displays much better quality than the previous (Figure
6C and 7C): we now see the protective nets in the slope as well as the blue road sign
announcing the tunnel. The vegetation is also observable and the tunnel entry is similarly
modelled as the 2016 GoPro point cloud (Figure 6D).
The SfM-MVS point cloud derived from GoPro images gives a significantly better
representation of the whole scene, especially on the top of the model. Slope structures and
protective nets are well modelled, but not the small vegetation. The comparison between the
2016 LiDAR scan (Figure 6E) and the three SfM-MVS with GSV images point clouds does
not allow to identify terrain deformation on the cliff. Moreover, the source area of the rockfall
is not observable from the GSV images because it is located higher in the slope, outside of the
images.
A great majority of points consistently displayed distances between the LiDAR scan mesh and
the SfM-MVS point clouds ranging between +/- 2 m (Figure 7 A-C). Protective nets degrade
the results because it generates badly modelled surfaces corresponding to the nets on some
cliff sections (such as the red-blue section on the top-right of the July 2014 cloud (Figure
7A)). Considering the tunnel entry (Figure 7 D-F) the average distance point clouds - LiDAR
mesh varies from -3 to -6 cm (depends mainly on the alignments of the clouds). Standard
deviations vary from 22 cm for the 2010 point cloud to 11 cm for the 2016 point cloud. On a
part of the wall above the tunnel (grey colour polygon on Figure 7 D-F), the average distance
point cloud - LiDAR mesh varies from -3 cm to -18 cm with standard deviations of 3 cm for
the 2010 point cloud, 4 cm for the 2014 point cloud et 6 cm for the 2016 point cloud (Figure 9
and Table 2). We observe again on this site that the improvement of the GSV camera
resolution and image quality improve the processing. The information on the pictures source,
date, point density and on the program used is given in Table 1.
A strong limiting factor on this site is the non-optimal camera locations. Indeed, the location
of the cliff above a tunnel portal does not allow for a lateral movement between the camera
positions with regard to the cliff. The maximal viewing angle (in blue colour on the Figure
2A) is about 35° compared to 170° for the site 1, and 115° for the site 2, that is 3 to 5 time
smaller than for the other studied sites.
4.4   Discussion
With the experience acquired during the research, we can highlight the following
recommendations to improve results of SfM-MVS with SVI images. (A) Firstly, the distance
between the image point of view and the subject and the size of the subject are important
because it influences the pixel size on the subject. In case study 1, the location of the cliff next
to the road (< 1 m) allows to get images with a good resolution for the studied object. In case
study 2, the area under investigation is too far from the road (500 – 1'400 m) and small
structures cannot be seen in the landslide. (B) Secondly, the ability to look at the scene from
different angles (Figure 2A) is a determining factor to obtain good results. The greater is this
"view angle", the better the results will be. Case study 1 with a view angle of almost 180° is

optimal because the object is observable from half a circle. View angle of case study 2 (115°) is enough to get many different views of the subject from different angles. The view angle is too narrow to have enough different point of view of the cliff on case study 3 (35°). (C) Thirdly, results are influenced by the image quality and especially by their exposition, contrast and type of sensor, which has progressively been improved during the last years. Image quality varies considerably on different images sets. Case study 1 is again the best study case in term of image quality. Both image sets have optimal solar exposition and shadows are not strong. Case study 2 has sets with very different images quality. Some sets are well exposed, others not. Clouds are present on few image sets. For case study 3, we have a lot of over- and underexposed images on behalf of the situation of the site (incised valley with a southwest oriented slope with a lot of light or shadow). The problem of images quality concerns Google too because it has removed from Google Maps very underexposed GSV images taken in August 2014 on site 3 at the end of 2016.

According to our findings, small landslides and rockfalls ($<0.5$ m$^3$) can be detected when the slope or the cliff is close to the road (0-10 m), as it was shown on site 1. Conversely, large slope movements and collapses ($>1$'000 m$^3$) can be detected when the studied area is far away from the road (up to 0.5-1 km) like on site 2. On such sites, small changes ($<1$ m$^3$) can correspond to either real rockfalls or errors resulting from processing like on the toe of almost all point 3D clouds of Séchilienne landslide (Figure 5 A2-H2). The measured differences between the point clouds on stable areas show interesting results once the point clouds alignment is well done. Thus, we observed standard deviations of few decimetre on stable areas on site 1 (Figure 3D), between 0.5 and 1.1 m on site 2 and between 11 and 22 m on the tunnel entry on site 3. Standard deviations increase on site 2 when point clouds are compared on their entire surface (Figure 5 A2-H2, Table 1). This is attributable to the occurrence of slope movements generating material increase or decrease and thereby, increasing standard deviations of the distance between the two compared point clouds. It can also be due to a bad 3D point cloud alignment. Indeed, cloud alignment is not always easy on some point clouds because of low point density, because of voids in the point clouds (like in the landslide toe in Figure 5 F2) and because of the roughness of the terrain. In such difficult alignment cases, it was tried to align the point clouds on stable parts where point density was high.

Our study highlighted important differences on 3D model reconstruction using different software, consistently with previous works (Micheletti et al., 2015; Gomez-Gutierrez et al., 2015, Niederheiser et al., 2016). Agisoft PhotoScan performed better than VisualSFM when

using both GSV images from Google Earth Pro (Figure 5F-G) and pictures acquired from a
GoPro Hero camera (Figure 5H). Nevertheless, VisualSfM performed better than Agisoft
PhotoScan on print screens captures from SVI. The only difference between these sources of
information is the resolution: 2.3 Mpx for print screens from Google Maps, 16.8 Mpx for
images saved from Google Earth Pro (and 12 Mpx for GoPro camera), stressing the
importance of picture resolution on the quality of the 3D model.
The point density was evaluated according to the distance between the image point of view
and the subject and the image types and processing software. The obtained results and the
derived trends indicate that the use of GSV images from Google Earth Pro with VisualSFM
increases by a factor two the point density compared to the processing of GSV print screens
with VisualSFM. The processing of GSV images from Google Earth Pro with Agisoft
PhotoScan increases by a factor ten the point density compared to the processing of GSV print
screens with VisualSFM (trend strips in Figure 8). The expected point density of the 3D point
clouds from GSV print screens processed in VisualSFM of a subject located few meters from
the camera ("Basse-Corniche" dots on Figure 8) is about 300 points/m$^2$, about 50 points/m$^2$
for an area located at about 100 m ("Arly" dots on Figure 8) and about 0.5 point/m$^2$ for an
area located at about 700 m ("Séchilienne" dots on Figure 8).

Despite the above mentioned prospects, some drawbacks were also observed. The main
limitation found in this study is that SfM-MVS processing is designed to retrieve the internal
orientation of standard cameras, whereas the images used in this research do not correspond to
a standard camera due the construction of the panoramas. Indeed, the main problem comes
from the different deformations on GSV print screens or images due to the panoramas
construction. Same radial deformations, that are stronger than common camera lens, on each
images, like on fisheyes images from GoPro cameras, can be processed without limitation
with SfM software like Agisoft PhotoScan. In addition, images from GSV are often over- or
underexposed (case study 3) and their resolution is low for distant subjects (cases study 2 and
3), making difficult to obtain results with few decimetric accuracy with these constraints.
Making zoomed print screens from GSV images do not allow increasing the SfM-MVS
process results (case study 2) due to a low images resolution. Finally, the spatial repartition of
SVI is often problematic because there are not enough images along the track path and
because the road path does not often allow obtaining an efficient strategy concerning the
camera positions around the studied area (case study 3). Accessing to original (RAW) images
together with valuable data of camera calibration would considerably help deriving 3D point
clouds from GSV using modern photogrammetric workflows.
A simple development to improve our proposed approach would be that Google add the *back*
*in time function* into the Google Earth Pro. In this case, it would be possible to save GSV
images from any proposed time period and to process those images with Agisoft PhotoScan
(Figure 5G) and thus to obtain better results than when using VisualSFM (Figure 5F).
Knowing that Google services and functionalities of Google Maps and Google Earth are
evolving over time, it is possible that SfM-MVS with GSV images will be more efficient and
easier in a near future.

## 5   Conclusions

In this study it was possible to detect and characterize small landslides and rockfalls ($<0.5$ m$^3$)
for study areas relatively close to the road (from 0 to 10 m); complementarily, it was possible
to detect large scale landslides or rock collapses ($>1$'000 m$^3$) over areas located far away from
the road (hundred meters or more). This information is of great interest when no other data of
the studied area has been obtained.
The proposed methodology provides interesting but challenging results due to some
constraints linked to the quality of the input imagery. The inconsistent image deformations
and the impossibility of extracting the original images from a street view provider are the
most important limitations for 3D model reconstruction derived from SVI. Following
constraints strongly limit the proposed approach: large distances between the camera position
and the subject of investigation, presence of obstacles between the studied area and the road,
image quality, poor meteorological conditions, non-optimal images repartition, reduced
number of images, existence of shadows/highlighted areas. The quality of the final product
was observed to be mainly dependent on the images quality and of the distance between the
studied area and image perspectives.
Although of the above mentioned limitations, SfM-MVS with SVI can be a useful tool in
geosciences to detect and quantify slope movements and displacements at an early stage of
the research by comparing datasets taken at different time series. The main interest of the
proposed approach is the possibility to use archival imagery and deriving 3D point clouds of
an area that has not been captured before the occurrence of a given event. This will allow
increasing database on rock slope failures, especially for slope changes along roads which
conditions are favourable for the proposed approach.

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

# 7 Figures

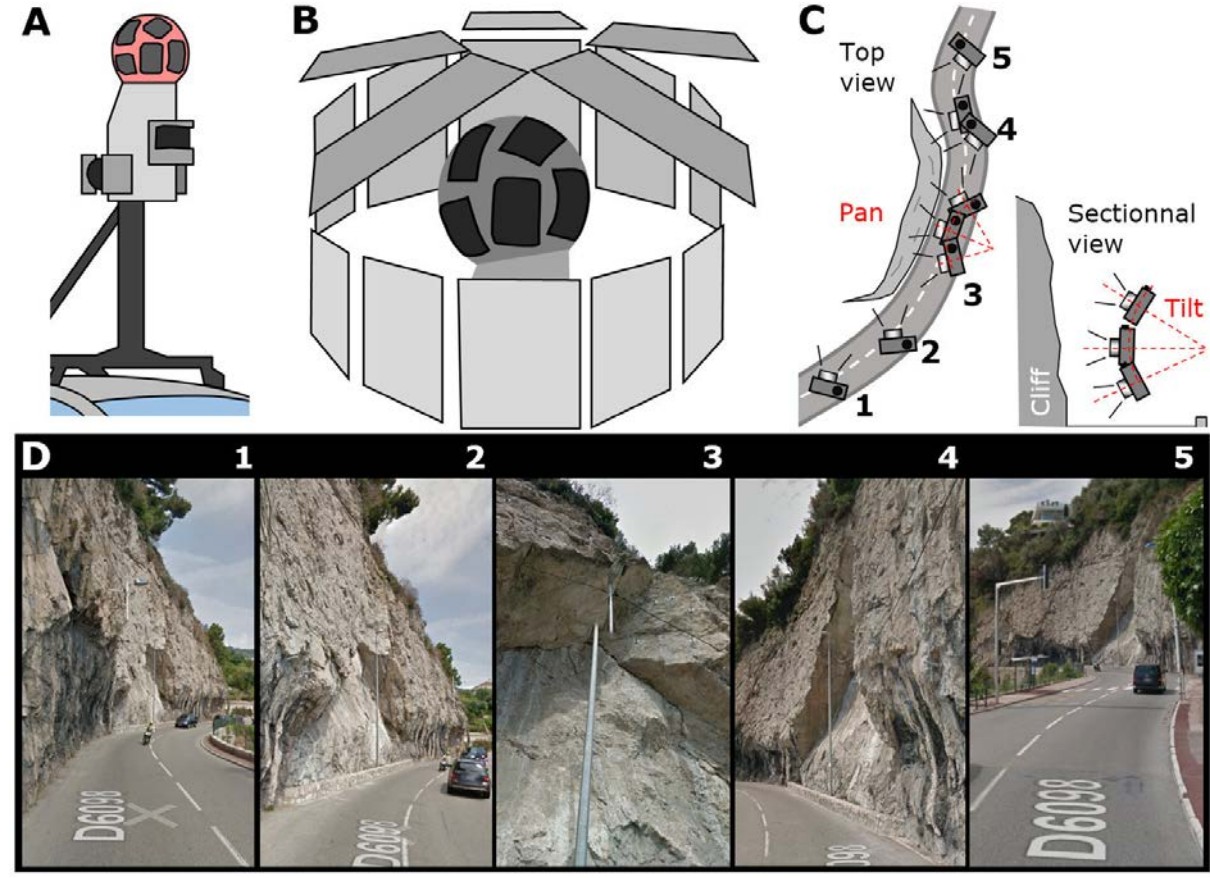


*Figure 1: Google Street View (GSV) imagery functioning. A: Schema of the GSV spherical camera system mounted on a car*
*roof. Sensors in black colour are LiDAR on which are draped the GSV images (based on Google Street View 2017). B:*
*Functioning of the GSV spherical panorama built with fifteen images. C: Strategy of the GSV service for SfM-MVS*
*photogrammetry. Numbers correspond schematically to the images in D. D: Screen captures of GSV photos from the study*
*site 1. The image numbers correspond to those in C. Note the gap on the street-lamp in images 3 due to the panorama*
*construction from the GSV pictures.*

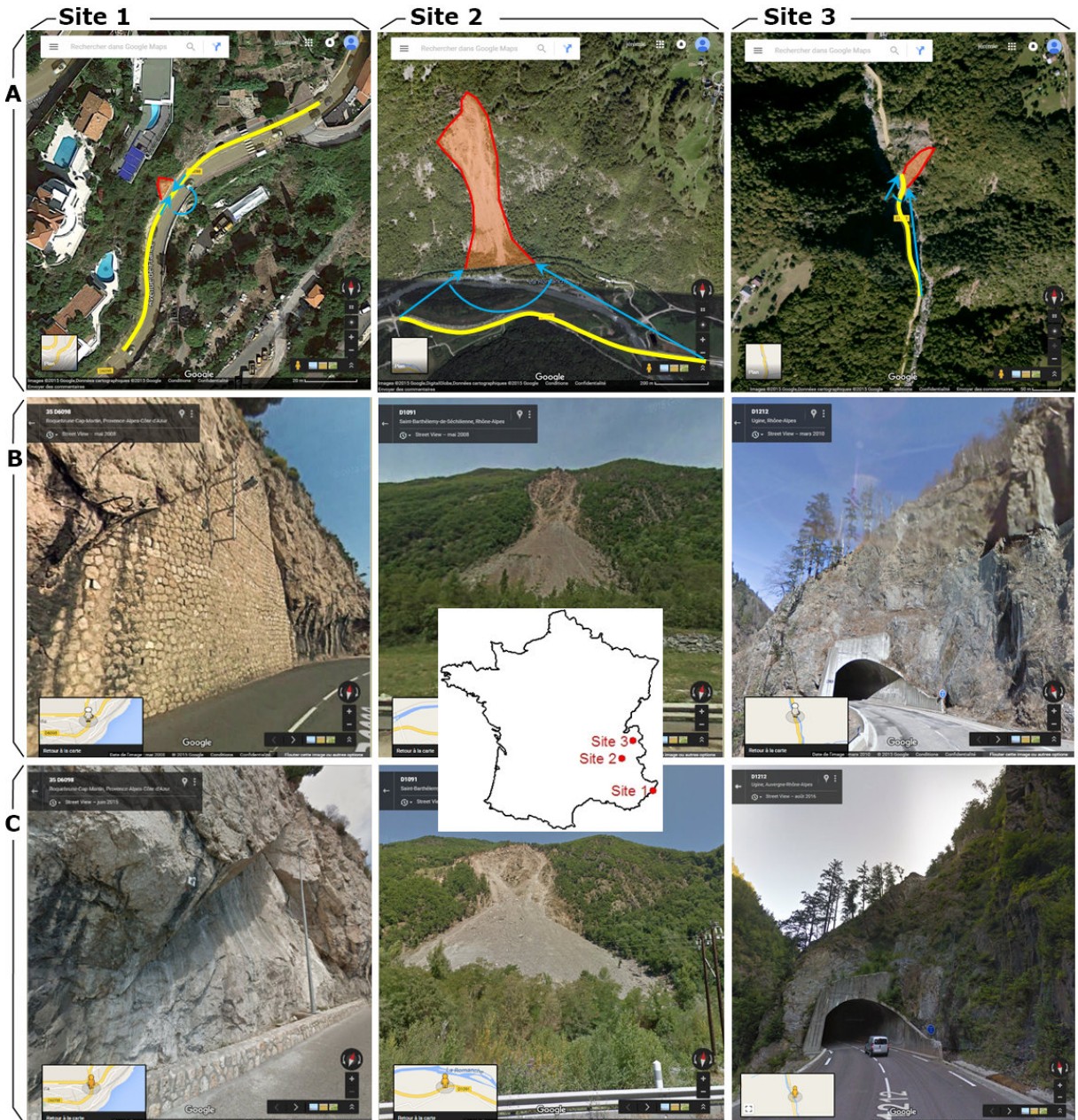


*Figure 2 : The three French studied sites (1: Basse-Corniche, 2: Séchilienne and 3: Arly gorges). A: Google Maps aerial view of the sites (in red) with the road path (yellow) used to take the GSV images of the scenes and the view angle (blue) of the images point of view around the sites. B: First GSV of the sites. C: Last GSV of the sites.*


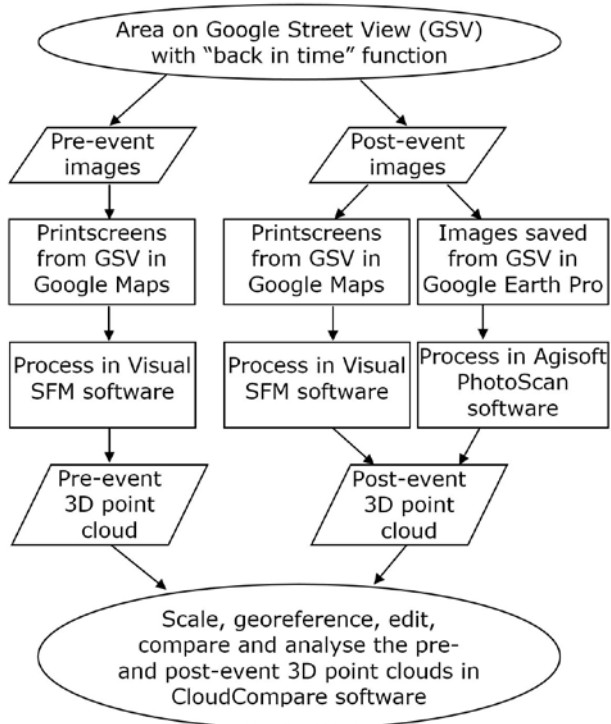


*Figure 3: Flowchart of the SfM-MVS processing with GSV images on an area with the "back in time" function available.*
*Pre-event images are displayed using the "back in time" function in GSV. Post-event images arise either from print screens*
*of GSV in Google Maps using or not the "back in time" function or from GSV images saved in Google Earth Pro. In this last*
*case, the last available proposed GSV images have a greater resolution as the print screens and can be processed in the*
*Agisoft PhotoScan.*

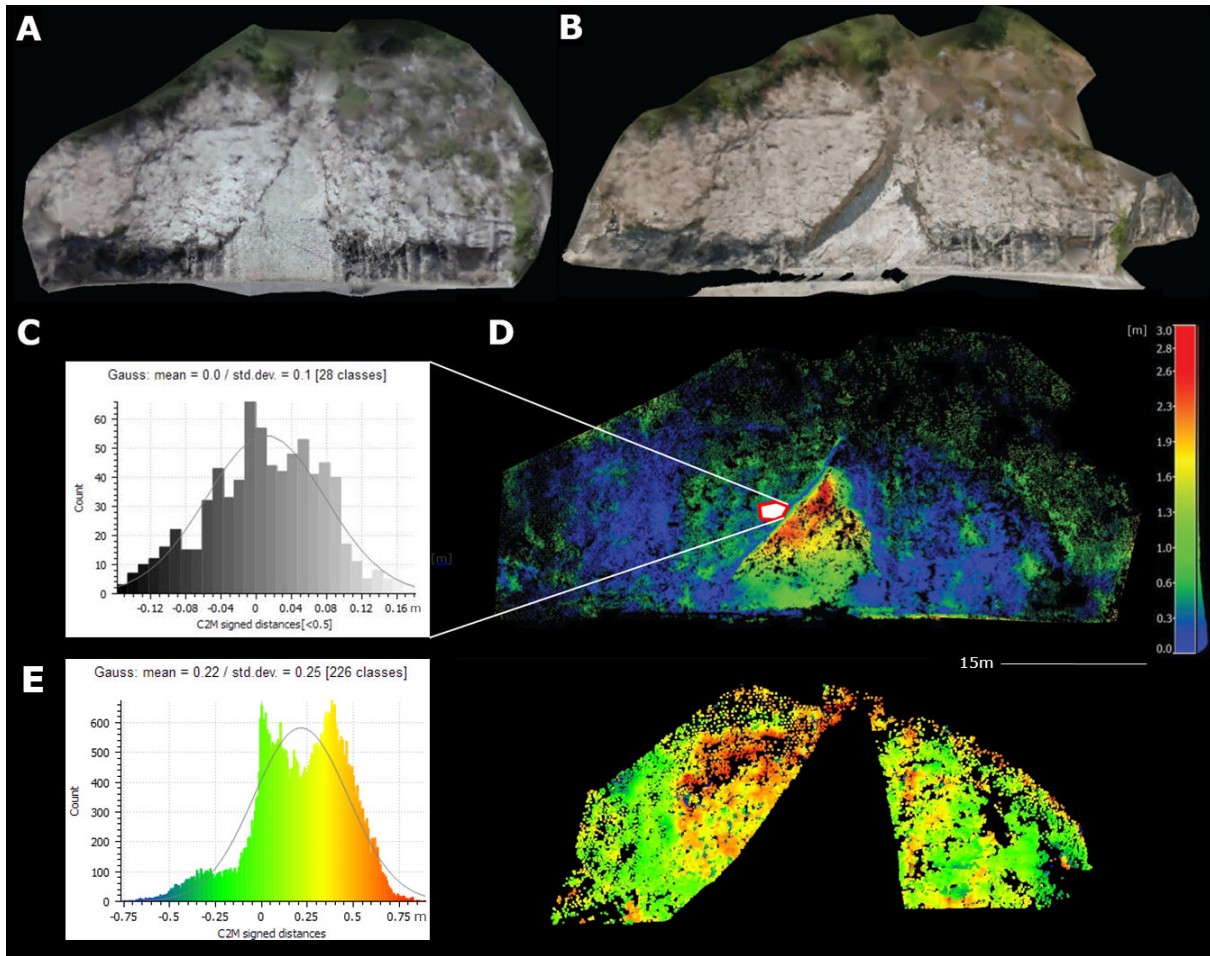


*Figure 4: Results at site 1 "Basse-Corniche". A: 3D model produced with GSV images taken before the event in 2008. B: 3D*
*model produced with GSV images taken after the event in 2014. C: Statistics on a small part of the wall (red colour polygon*
*on figure D) of 7'510 points between the two point clouds with the point-to-mesh strategy in the CloudCompare. D:*
*Comparison of the two point clouds of 2008 and 2014 on the entire surface of the 3D point clouds. The maximal horizontal*
*depth of the cliff is about 3.9 m. E: Comparison of the two point clouds of 2008 and 2014 on the entire stable parts of the cliff*
*(i.e. without vegetation) by not taking into account the collapsed wall (black triangle in the centre of the point clouds. The*
*information on the pictures source, date, point density and on the program used is given in Tables 1 and 2.*

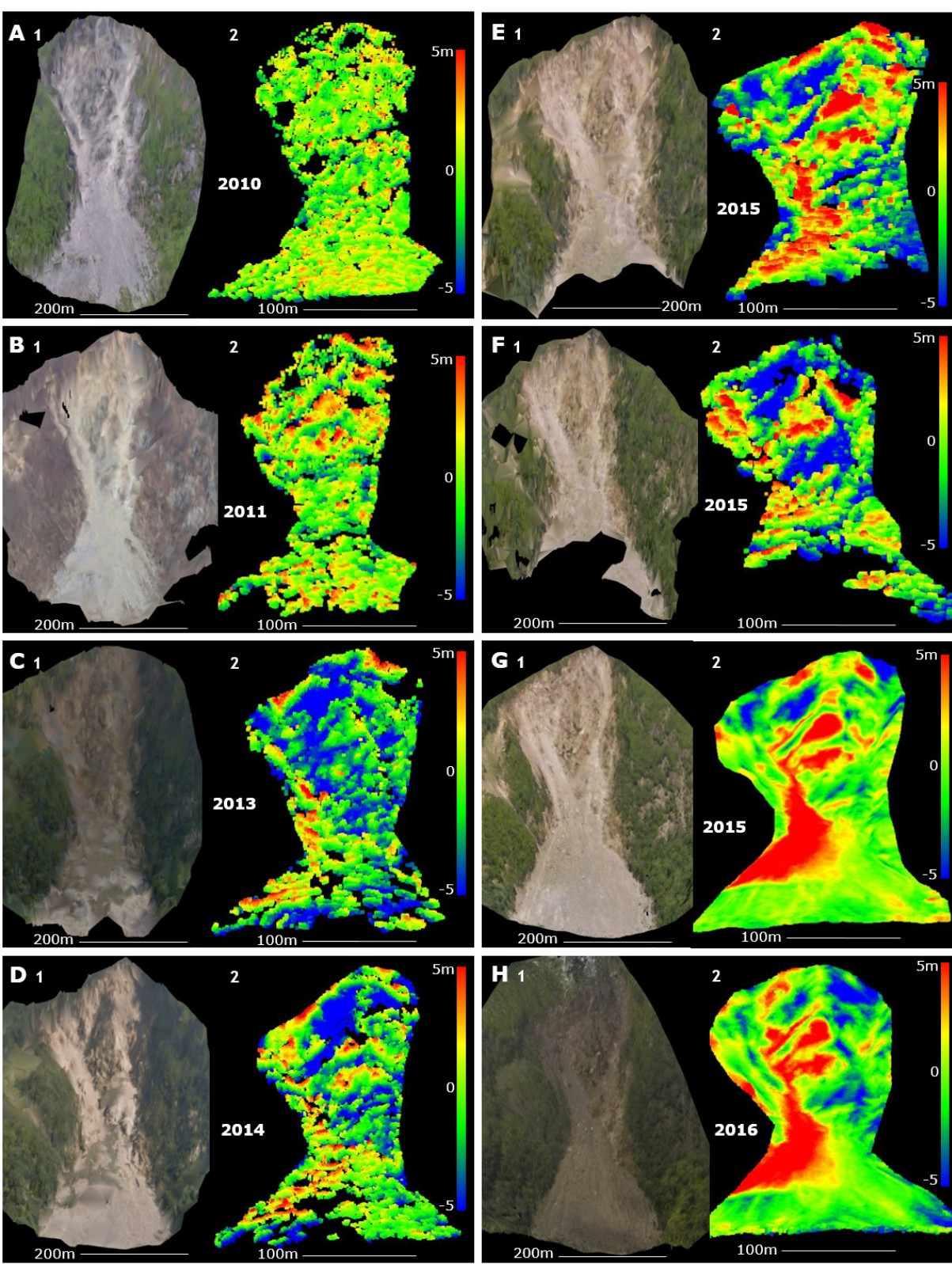

Figure 5 : Results at site 2 "Séchilienne". Eight points clouds from different images sets taken at six different time with three different image sources and processed with two different programs. Figures A1-H1: Meshs resulting from the respective point clouds. Figures A2-H2: point clouds comparison with a 50 cm LiDAR DEM from 2010 (red colour points is material increase; blue colour points are material decrease from the 2010 LiDAR cloud) with the point-to-mesh strategy in CloudCompare. The information on the pictures source, date, point density and on the program used is given in Table 1.

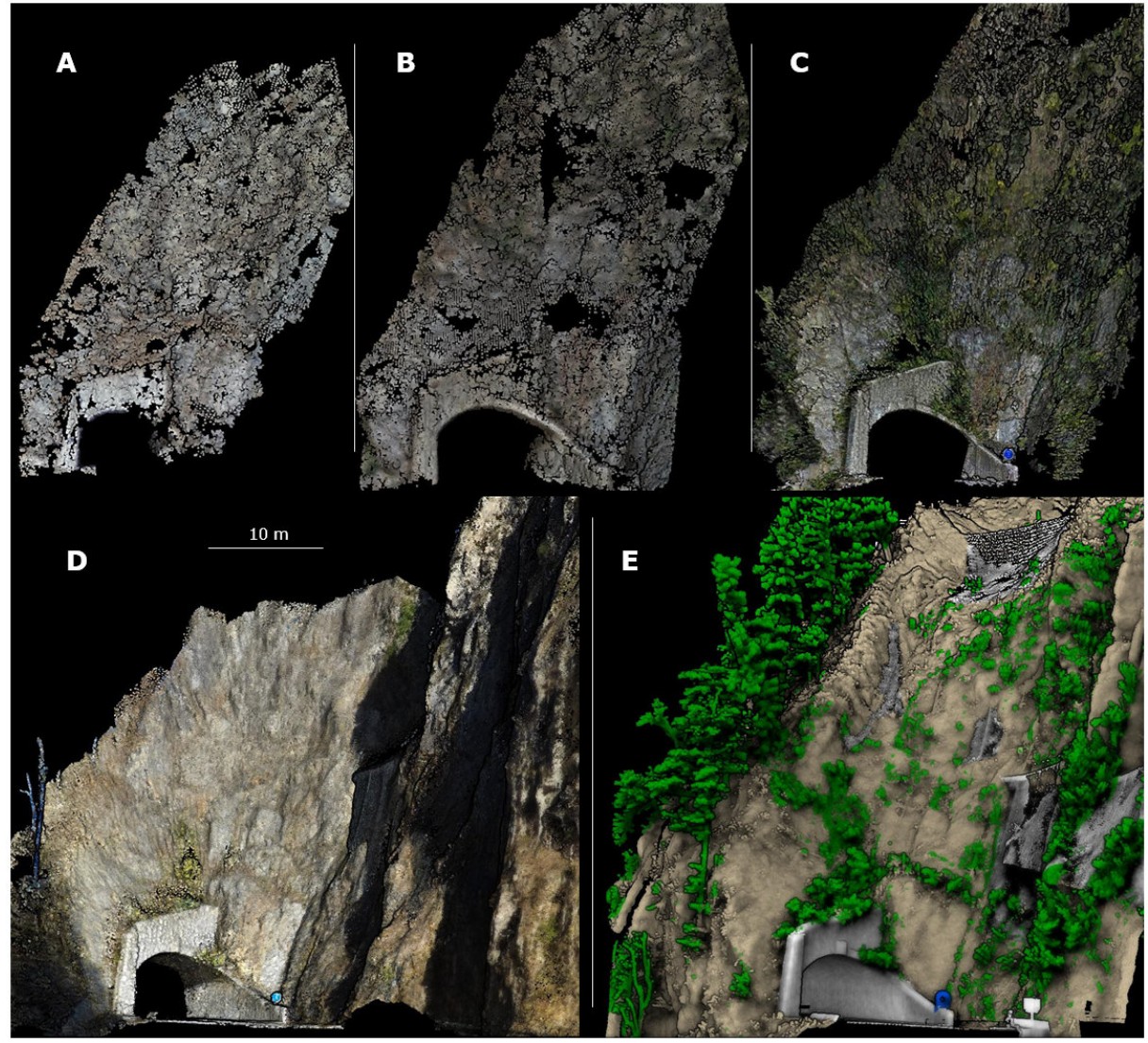


*Figure 6 : Results at site 3 "Arly gorges". Five points clouds from four different images sets sources and processed with two*
*different softwares and one LiDAR scan. A: March 2010 point cloud. B: July 2014 point cloud. C: August 2016 point cloud.*
*D: December 2016 point cloud taken on foot with a GoPro camera. E: December 2016 LiDAR cloud from an assembly of six*
*Optech terrestrial LiDAR scans. The grey elements in the cliff are the protective nets.*

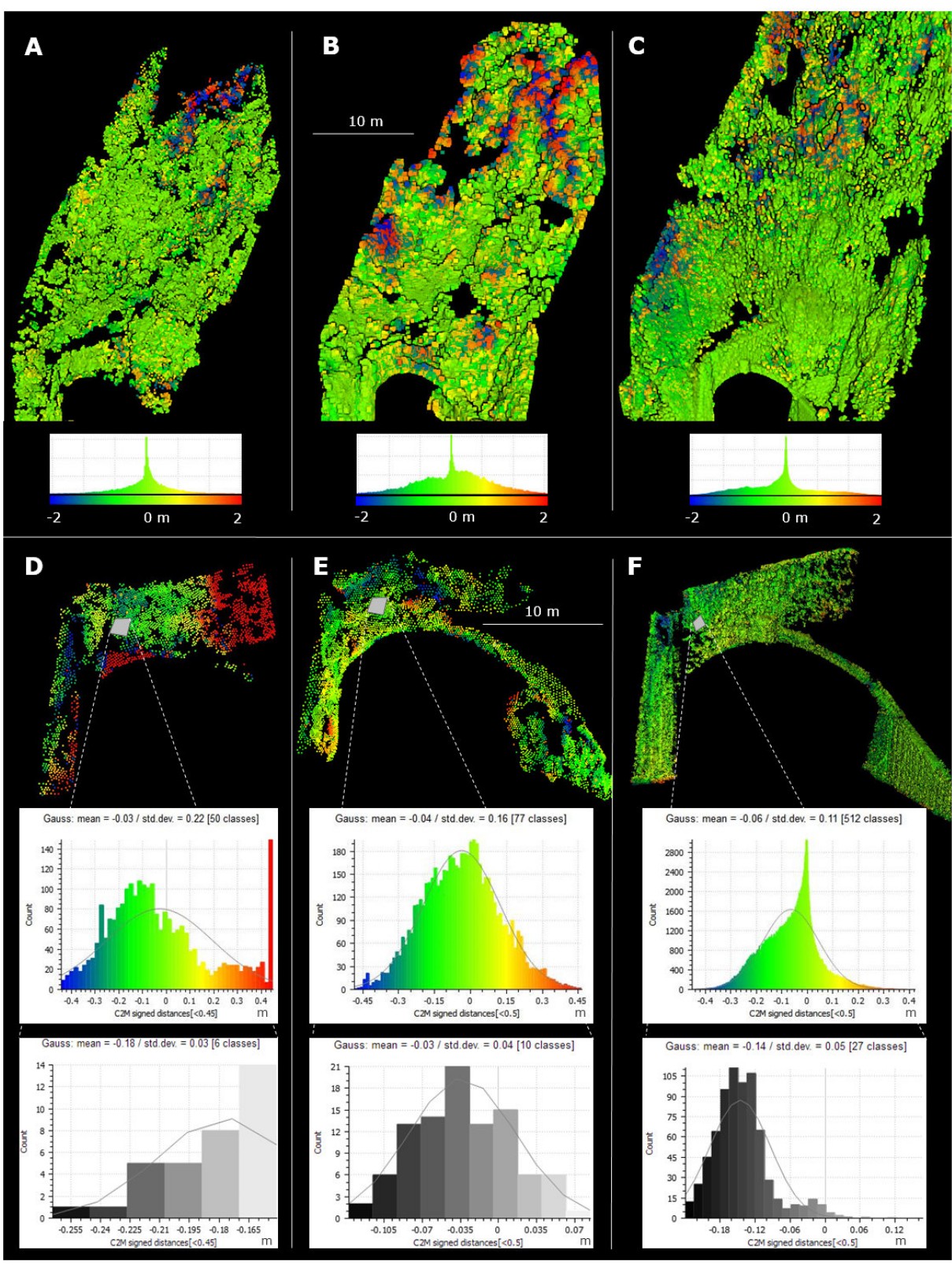


Figure 7: A-B-C: March 2010, July 2014 and August 2016 point clouds compared with December 2016 LiDAR DEM (red
colour points is material increase; blue colour points are material decrease from the 2016 LiDAR cloud) with the point-to-
mesh strategy on the CloudCompare. D, E, F: tunnel entry and part of the wall overlooking the tunnel (grey colour polygon)
of the March 2010, July 2014 and August 2016 point clouds compared with December 2016 LiDAR DEM. The information
on the pictures source, date, point density and on the program used is given in Tables 1 and 2.


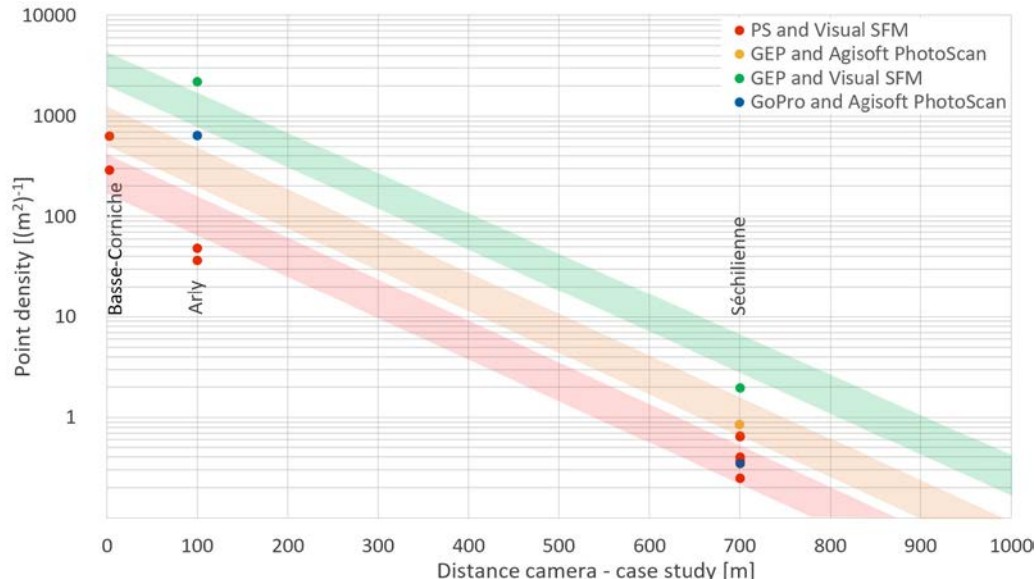


*Figure 8: Correlation between distance camera - case studies and the expected density of points from the three case studies.*
*The red colour dots are results of the three case studies point clouds obtained from Google Street View (GSV) print screens*
*(PS) in Google Maps (GM) processed with VisualSFM. The red strip represents the corresponding trend based on a negative*
*exponential function. The orange colour dot is the result of the Séchilienne point cloud obtained from GSV images saved in*
*Google Earth Pro (GEP) processed with VisualSFM. The orange strip represents the corresponding trend based on a*
*negative exponential function. The green colour dots are results of the Séchilienne and Arly point clouds obtained from GSV*
*images saved in (GEP) processed with Agisoft PhotoScan. The green strip represents the corresponding trend based on a*
*negative exponential function. By way of comparison, the blue colour dots represent the result of the Séchilienne and Arly*
*point clouds obtained with GoPro action camera images taken on the field and processed with Agisoft PhotoScan.*
*Figure 9: Correlation between distance camera - case studies and the expected standard deviation from the three case*
*studies. The dots are results of point clouds comparisons on the entire point cloud areas (Table 1). The triangle are results of*
*point clouds comparisons on partial point cloud area (Table 2). The red colour dots and triangle are results of the three case*
*studies point clouds obtained from Google Street View (GSV) print screens (PS) in Google Maps (GM) processed with*
*VisualSFM compared on the entire area. The orange colour dot is the result of the Séchilienne point cloud obtained from*
*GSV images saved in Google Earth Pro (GEP) processed with VisualSFM. The green colour dots and triangles are results of*
*the Séchilienne and Arly point clouds obtained from GSV images saved in (GEP) processed with Agisoft PhotoScan. By way*
*of comparison, the blue colour dots represent the result of the Séchilienne and Arly point clouds obtained with GoPro action*
*camera images taken on the field and processed with Agisoft PhotoScan.*

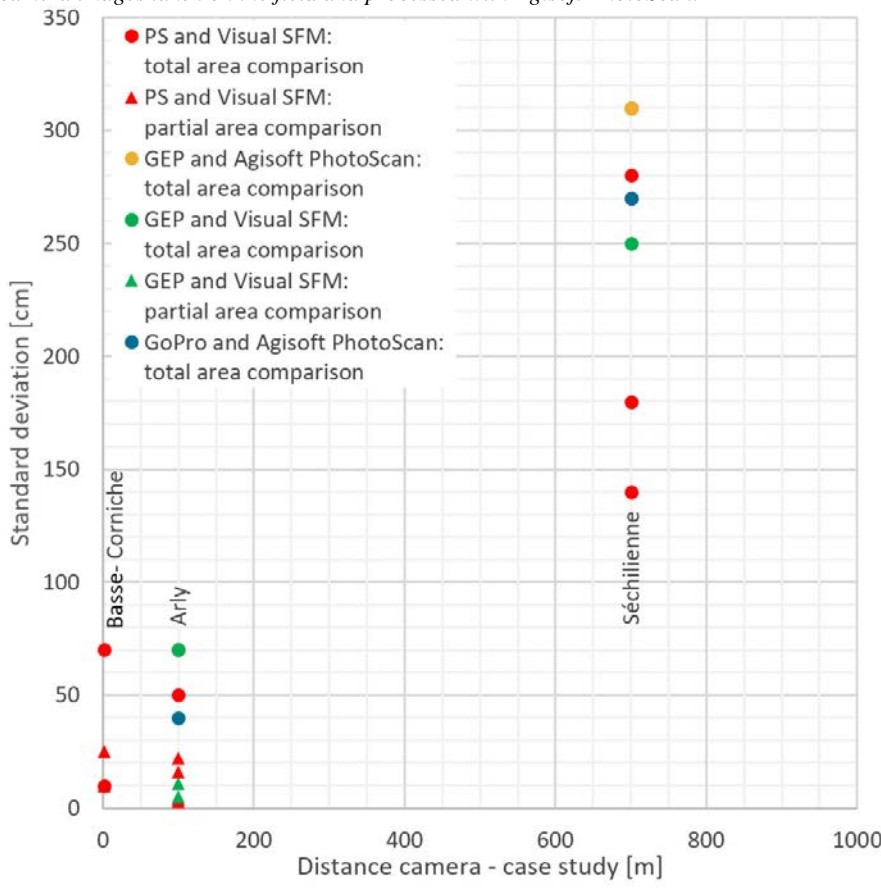



*Table 1: List of the fourteen point clouds presented in this paper.*

| Site | Figure | Date | Images source | Images size [pixel] | Images number | Point density[1] (pts/m$^2$) | Processing software | Number of points | Comparison | | |
|---|---|---|---|---|---|---|---|---|---|---|---|
| | | | | | | | | | With | Mean distance[2] [m] | Std. dev. [m] |
| Site 1 | Fig. 4A | 2008.05 | PS GSV from GM[3] | 1920 x 1200 | 60 | 290 | VisualSFM | 150'000 | )14.06[7] | 0.2 | 0.7 |
| | Fig. 4B | 2014.06 | PS GSV from GM[3] | 1920 x 1200 | 50 | 640 | VisualSFM | 182'000 | )8.05[8] | 0.0 | 0.1 |
| Site 2 | Fig. 5A | 2010.04 | PS GSV from GM[3] | 1920 x 1200 | 54 | 0.40 | VisualSFM | 18'000 | )AR[9] | -0.2 | 1.4 |
| | Fig. 5B | 2011.03 | PS GSV from GM[3] | 1920 x 1200 | 52 | 0.25 | VisualSFM | 9'500 | )AR[9] | -0.1 | 1.8 |
| | Fig. 5C | 2013.05 | PS GSV from GM[3] | 1920 x 1200 | 45 | 0.37 | VisualSFM | 12'500 | )AR[9] | -2.1 | 2.7 |
| | Fig. 5D | 2014.06 | PS GSV from GM[3] | 1920 x 1200 | 52 | 0.66 | VisualSFM | 25'000 | )AR[9] | -1.5 | 2.8 |
| | Fig. 5E | 2015.06 | PS GSV from GM[3] | 1920 x 1200 | 62 | 0.64 | VisualSFM | 23'500 | )AR[9] | -0.9 | 3.1 |
| | Fig. 5F | 2015.06 | GSV from GEP[4] | 4800 x 3500 | 80 | 0.86 | VisualSFM | 22'500 | )AR[9] | -1.7 | 3.1 |
| | Fig. 5G | 2015.06 | GSV from GEP[3] | 4800 x 3500 | 80 | 1.99 | Agisoft PhotoScan | 236'000 | )AR[9] | 0.6 | 2.5 |
| | Fig. 5H | 2016.05 | GoPro[5] | 4000 x 3000 | 75 | 0.35 | Agisoft PhotoScan | 46'000 | )AR[9] | -0.2 | 2.7 |
| Site 3 | Figs. 6A, 7A | 2010.03 | PS GSV from GM[3] | 1920 x 1200 | 66 | 40 | VisualSFM | 35'000 | )AR[10] | 0.0 | 0.5 |
| | Figs. 6B, 7B | 2014.07 | PS GSV from GM[3] | 1920 x 1200 | 111 | 50 | VisualSFM | 53'000 | )AR[10] | 0.1 | 0.7 |
| | Figs. 6C, 7C | 2016.08 | GSV from GEP[2] | 4800 x 3107 | 64 | 2200 | Agisoft PhotoScan | 3'1850'000 | )AR[10] | -0.1 | 0.7 |
| | Fig. 6D | 2016.12 | GoPro[6] | 4000 x 3000 | 50 | 650 | Agisoft PhotoScan | 2'217'000 | )AR[10] | 0 | 0.4 |


[1] Point density around a search radius of 2 m.
[2] Average distance between the mesh of the reference point cloud and the compared point cloud using the point-to-mesh strategy.
[3] Print screens (PS) of Google Street View (GSV) from Google Maps (GM).
[4] Google Street View (GSV) images saved in Google Earth Pro (GEP).
[5] GoPro Hero4+.
[6] GoPro Hero5 Black with GNSS chip integrated.
[7] Comparison between the entire point clouds of May 2008 and June 2014 (Figure 3D).
[8] Comparison of a small cliff area of the May 2008 and June 2014 point clouds (Figure 3C).
[9] Comparison with the 50 cm airborne LiDAR DEM from 2010.
[10] Comparison with the December 2016 LiDAR DEM (6'930'000 points) without vegetation from an assembly of six Optech terrestrial LiDAR clouds.

*Table 2: List of the eight partial point cloud comparisons.*

| Site | Figure | Date | Images source | Images size [pixel] | Processing software | Comparison | | | |
|------|--------|------|---------------|--------------------|--------------------|-----------------|---------|---------------------------|----------------|
| | | | | | | Comparative area | With | Mean distance[1] [cm] | Std. dev. [cm] |
| Site 1 | Fig. 4C | 2008.05 | PS GSV from GM[2] | 1920 x 1200 | VisualSFM | Small cliff part | 4.06[4] | 0 | 10 |
| | Fig. 4E | 2008.05 | PS GSV from GM[2] | 1920 x 1200 | VisualSFM | Entire cliff without wall and vegetation | 4.06[4] | 22 | 25 |
| Site 3 | Fig. 7D 1 | 2010.03 | PS GSV from GM[2] | 1920 x 1200 | VisualSFM | Tunnel entry | )AR[5] | -3 | 22 |
| | Fig. 7D 2 | 2010.03 | PS GSV from GM[2] | 1920 x 1200 | VisualSFM | Small part of tunnel entry | )AR[5] | -18 | 3 |
| | Fig. 7E 1 | 2014.07 | PS GSV from GM[2] | 1920 x 1200 | VisualSFM | Tunnel entry | )AR[5] | -4 | 16 |
| | Fig. 7E 2 | 2014.07 | PS GSV from GM[2] | 1920 x 1200 | VisualSFM | Small part of tunnel entry | )AR[5] | -3 | 4 |
| | Fig. 7F 1 | 2016.08 | GSV from GEP[3] | 4800 x 3107 | Agisoft PhotoScan | Tunnel entry | )AR[5] | -6 | 11 |
| | Fig. 7F 2 | 2016.08 | GSV from GEP[3] | 4800 x 3107 | Agisoft PhotoScan | Small part of tunnel entry | )AR[5] | -14 | 5 |

[1] Average distance between the mesh of the reference point cloud and the compared point cloud using the point-to-mesh strategy.
[2] Print screens (PS) of Google Street View (GSV) from Google Maps (GM).
[3] Google Street View (GSV) images saved in Google Earth Pro (GEP).
[4] Comparison between the entire point clouds of May 2008 and June 2014 (Figure 3D).
[5] Comparison of a small cliff area of the May 2008 and June 2014 point clouds (Figure 3C).
[6] Comparison with the December 2016 LiDAR DEM (6'930'000 points) without vegetation from an assembly of six Optech terrestrial LiDAR clouds.