# Peer review of "Using street view imagery for 3D survey of rock slope failures"

_Natural Hazards and Earth System Sciences, 2017_

## Referee Comment (RC1) · Dr. Lato (Referee) · 30 Jun 2017

Overall this is an interesting paper, but i think it requires some more scientific thought and the quality of the analysis and figures require improvement. As is the paper reads like a conference paper, not a journal paper.

In general it is an interesting idea, but the tests are limited to three sites with dramatically different settings. This limits the authors ability to quantify the method, they should have focused on a slope type (close and rock, or far and soil) and tested three or four of that type. This would have led to a more robust analysis and conclusion. As it stands the authors state it works in some places better than others based on picture quality, lightening, etc. These are not geotechnical qualities, which should have been the

focus. If the focus was on image quality, NHESS is the wrong journal for submission.

I encourage the authors to dive deeper into their work and test many more sites and resubmit.

Some specific comments: Stating LiDAR is expensive and demanding from a logistics point of view is irrelevant, especially when referencing a paper from 2014, that was likely written in 2012 or 2013. Modern applications of lidar are neither of those. Avoid general language with little meaning like "reasonably good" You state in Section 4 VisualSFM gave the 'best results' – this is arbitrary, you need numbers to back this up. What metric are you using to define 'best'? Section 4.1: Standard deviation of the error below 20 cm – what error are you assessing? 3D vector, Z, or XY? Your volume estimates do not have ranges, yet your point cloud has alignment errors. You should report volumes with +/- amounts. Again, 'reasonably good' should not be used in a scientific paper. Same for 'We hardly perceive' 'Same strong radial' ? In your conclusions you state the method is useful to 'quantify slope movements and displacements' yet you did not show this anywhere in your paper. You showed the ability to measure failed volumes, not displacements. This is a misleading conclusion. On your change mapping images the colours below the limit of detectable change should be coloured grey. All figures need a scale bar. Figure caption 5 is too long. The min and max difference calculated in Table 1 adds no value, those points are likely outliers.

---

## Referee Comment (RC2) · Olga Mavrouli (Referee) · 4 Sep 2017

This is a very interesting paper with very useful and innovative ideas and I believe that research towards this direction is promising. However, reading the manuscript I missed a strong and solid part on technical specifications for the methodology that is used and for the quantitative analysis of the results, which is the core and the added value of this work. In that sense, I suggest to the authors, to enrich and support the description of the methodology, providing detailed information on the processes followed and to present a more thorough and detailed analysis of their results, in quantitative terms.

Some specific comments can be found at the attached .pdf file.

[Figure]

Please also note the supplement to this comment:
https://www.nat-hazards-earth-syst-sci-discuss.net/nhess-2017-49/nhess-2017-49-RC2-supplement.pdf

─────────────────────────────

**Supplement:**

[revised manuscript text omitted]

---

## Author Comment (AC2) · 4 Oct 2017

Acknowledgements: Second author was funded by the H2020 Program of the European Commission: Marie Skłodowska-Curie Individual Fellowships [MSCA-IF-2015-705215].

---

## Author Response (AR1)

**Response to the reviews**

**In bolt: text added**
**Referee 1**

Overall this is an interesting paper, but i think it requires some more scientific thought and the quality of the analysis and figures require improvement. As is the paper reads like a conference paper, not a journal paper. In general it is an interesting idea, but the tests are limited to three sites with dramatically different settings. This limits the authors ability to quantify the method, they should have focused on a slope type (close and rock, or far and soil) and tested three or four of that type. This would have led to a more robust analysis and conclusion. As it stands the authors state it works in some places better than others based on picture quality, lightening, etc. These are not geotechnical qualities, which should have been the focus. If the focus was on image quality, NHESS is the wrong journal for submission. I encourage the authors to dive deeper into their work and test many more sites and resubmit. Some specific comments: Stating LiDAR is expensive and demanding from a logistics point of view is irrelevant, especially when referencing a paper from 2014, that was likely written in 2012 or 2013. Modern applications of lidar are neither of those. Avoid general language with little meaning like "reasonably good" You state in Section 4 VisualSFM gave the 'best results' – this is arbitrary, you need numbers to back this up. What metric are you using to define 'best'? Section 4.1: Standard deviation of the error below 20 cm – what error are you assessing? 3D vector, Z, or XY? Your volume estimates do not have ranges, yet your point cloud has alignment errors. You should report volumes with +/- amounts. Again, 'reasonably good' should not be used in a scientific paper. Same for 'We hardly perceive' 'Same strong radial' ? In your conclusions you state the method is useful to 'quantify slope movements and displacements' yet you did not show this anywhere in your paper. You showed the ability to measure failed volumes, not displacements. This is a misleading conclusion. On your change mapping images the colours below the limit of detectable change should be coloured grey. All figures need a scale bar. Figure caption 5 is too long. The min and max difference calculated in Table 1 adds no value, those points are likely outliers.

Comment: In general it is an interesting idea, but the tests are limited to three sites with dramatically different settings. This limits the authors ability to quantify the method, they should have focused on a slope type (close and rock, or far and soil) and tested three or four of that type.
*Answer: The idea behind the three different sites is to demonstrate the capacity of the method to work on different topographic areas with different slope types with different distances image point of view – site. The first site (Monaco) shows the modelling of an anthropic slope with a wall collapse. The danger of wall collapse on a transportation track can be found everywhere around the world. We find that this case study is pertinent because it is representative of a real danger for transportation networks. Site 2*

*(Séchilienne) shows the capacity of the proposed method to model a large landslide away from the road. With 6 different time steps, results shows a slope evolution over the years which corresponds to the surface changes measured with LiDAR scans. The accuracy is obviously lower as the LiDAR accuracy, but it allows to observe the main surface change. The third site (Arly) focus on a steep slope threatening a road tunnel entry. A rockfall occurred already on this area and protective measures have been built. This site shows the limit of the method in the vertical axis because some images were taken close to the cliff which is much higher than the Monaco wall. We believe that the three sites on different slopes and different settings shows the capacity and limits of the method which can be deployed on several topographic situations.*

Comment: As it stands the authors state it works in some places better than others based on picture quality, lightening, etc. These are not geotechnical qualities, which should have been the focus. If the focus was on image quality, NHESS is the wrong journal for submission. I encourage the authors to dive deeper into their work and test many more sites and resubmit.

Answer: *The manuscript presents a uncommon free method that obtain 3D point cloud of a slope without field visit. The focus is clearly not on image quality, but image quality must be mentioned as it is an important condition to obtain results. This is why those "no geotechnical qualities" are mentioned. With the manuscript improvement (see below), it focuses now more on the method (with an added flowchart) and its results.*

Comment: Some specific comments: Stating LiDAR is expensive and demanding from a logistics point of view is irrelevant, especially when referencing a paper from 2014, that was likely written in 2012 or 2013. Modern applications of lidar are neither of those

Answer: *Although LiDAR references are indeed not so actual, we still maintain that LiDAR, compared to the proposed method (free, any field work), is still more expensive and more demanding from a logistics point of view (except handle LiDAR like GeoSlam for the logistic point of view). Scanning cliffs of the case studies demand few hours of field work (as we made on site 2 and site 3).*

Comment: Avoid general language with little meaning like "reasonably good" You state in Section 4 VisualSFM gave the 'best results' – this is arbitrary, you need numbers to back this up. What metric are you using to define 'best'?

Answer: *We totally agree with this remark. We try to define our magnitude order assessment with values or examples. We have for example now: "**This accuracy allows to detect object of tens cetimeters size**" still "reasonably good results". "Best results" terms have been deleted.*

Comment: Section 4.1: **Standard deviation** of the error below 20 cm – what error are you assessing? 3D vector, Z, or XY? Your volume estimates do not have ranges, yet your point cloud has alignment errors.

Answer: *"Error" is a wrong term. It is a distance between a mesh and a point cloud. The computed distance is a 3D vector from the mesh triangle to the cloud point. The sentence is now: "***The computed shortest distance, in signed values, between the mesh and the point cloud is a 3D vector from the mesh triangle to the 3D point.***" (Page 4, line 29)*

*https://www.geometrictools.com/Documentation/DistancePoint3Triangle3.pdf*

*https://tel.archives-ouvertes.fr/file/index/docid/500182/filename/manuscrit_19052006_electronic.pdf*

*(Page 36, Section 2.2.1, Figure 2.1 of the linked document)*

Comment: In your conclusions you state the method is useful to 'quantify slope movements and displacements' yet you did not show this anywhere in your paper. You showed the ability to measure failed volumes, not displacements. This is a misleading conclusion.

Answer: *Right, "displacement" is term a little bit too optimistic. It is possible to detect displacement in specific cases (displacement of few meters between the image sets, 3D point cloud with a accuracy of few decimetres, etc.) but we have replaced the term "displacement" with "surface changes". Surface change on site 1 is the wall collapse, on site 2 it is the rockfall deposit and the rockfall scare, on site 3 there is no surface change because the landslide is located on a cliff part not visible with the GSV images.*

Comment: On your change mapping images the colours below the limit of detectable change should be coloured grey. All figures need a scale bar. Figure caption 5 is too long.

Answer: *All figures have a scale bar. Figure 5 is cut into 2 different figures with 2 captions.*

Comment: The min and max difference calculated in Table 1 adds no value, those points are likely outliers.

Answer: *Right, the min and max differences were deleted because their contribution was not very interesting for this manuscript. Point density of the 3D point cloud replaces those deleted values in the Table 1.*

**Referee 2**

This is a very interesting paper with very useful and innovative ideas and I believe that research towards this direction is promising. However, reading the manuscript I missed a strong and solid part on technical specifications for the methodology that is used and for the quantitative analysis of the results, which is the core and the added value of this work. In that sense, I suggest to the authors, to enrich and support the description of the methodology, providing detailed information on the processes followed and to present a more thorough and detailed analysis of their results, in quantitative terms.

Answer: "Methodology" (#3) and "Results and discussion" (#4) sections have been significantly rewritten. Table 2 has been added, as Figures 3, 8 and 9.

*Page 3: Lines 19-20*
If these parameters are not known beforehand, three pictures is the minimum requirement (Westoby 2012), **and** about six pictures is preferred.
*Answer: "And" added.*

*Page 4: Lines 16-17*
We used two image sets from for the first study site,  **eight** images sets for the second study site and four image sets for the third study site, with dates ranging from May 2008 up to December 2016, as described in table 1.
*Answer: "Height" replaced by "eight".*

*Page 4: Lines 26-29*
To perform temporal comparisons on each site, images were taken at the different dates proposed by GSV. We used the SfM-MVS programs VisualSFM (Wu 2011) and Agisoft PhotoScan (Agisoft 2015) for dense point cloud reconstruction and CloudCompare (Girardeau-Montaut 2011) for point cloud visualization and comparison. Comparison between two point clouds was made using point-to-mesh strategy.
*Question: It would be interesting to explain here, how the scaling and georeferencing was done, if you used control points and how many of them.*
*Answer: Sentence replaced by:* "To perform temporal comparisons on each site, images were taken at the different dates proposed by GSV **with pre- and post-event images sets**. We used the SfM-MVS program VisualSFM (Wu 2011) **for dense point cloud reconstruction for the print screens images from Google Maps** and we used CloudCompare (Girardeau-Montaut 2011) software for point cloud visualization and comparison. Comparison between two point clouds was made using point-to-mesh strategy."

*Further, sentence added:* "**The rough scaling and georeferencing of the obtained 3D point clouds were been made without ground control points but only with coordinates of few points extracted from Google Maps or French geoportal (Géoportail, 2016).**"

*Page 5: Line 1*

from print screens

*Question: what is the resolution of the images print screen? could you please provide some more technical information on the process and the result of the print screen? Are there certain specifications in order to achieve the result that you mention here?*

Answer: *Sentence replaced by:* "Beside the images taken from print screens as described above, we also obtained GSV images (**4800 x 3500 pixels, 16.8 Mpx**) from Google Earth Pro on sites 2 and 3 with the "save image" function." *for resolution information.*

*Further, sentence modified*: "In addition, GoPro Hero4+ images from a moving vehicle on the road were taken by the authors on site 2, as well a series of images captured using a GoPro Hero5 Black camera standing on site 3 (**image resolution of 4000 x 3000 pixels, 12 Mpx**)."

*Following sentence modified and replaced by (about the process):* "**This second way to get GSV allows to obtain images with a higher resolution as print screen images**. Unfortunately, there is no timeline function in this program and it is only possible to save Google Earth Pro images from the last picture acquisition, **i.e. generally post-event images. GSV images from Google Earth Pro were processed with the Agisoft PhotoScan (Agisoft 2015) software for dense point cloud reconstruction. The reason why we chose Visual SFM software to process GSV print screens images from Google Maps is because the processing of those print screens with Agisoft PhotoScan software is not possible while results of GSV images processing from Google Earth Pro is clearly better with Agisoft. The flowchart of SfM-MVS with GSV images combines also two image types from two different sources (print screens and saved images) processed into two softwares (Figure 3).**"

*Figure 3 added (process flowchart):* "**Flowchart of the SfM-MVS processing with GSV images on an area with the "back in time" function available. Pre-event images are print screens of GSV in Google Maps. Those GSV images are displayed using the "back in time" function in GSV and are processed in Visual SFM software. Post-event images arise either from print screens of GSV in Google Maps using or not the "back in time" function or from GSV images saved in Google Earth Pro. In this last case, the last available proposed GSV images have a greater resolution as the print screens and can be processed in the Agisoft PhotoScan software.**" (*Figure 3 caption).*

*Page 5: Line 12*

This information was used for quality assessment purposes.

*Question: It would be useful here, to get some information on the resolution of the images in each case.*

*Answer: Image resolution is now given (please see previous question).*

*Page 5: Lines 9-17*

Different results are obtained as a function on the software used for SfM-MVS processing. VisualSFM gave the best results with print screens from GSV while Agisoft PhotoScan could not align any GSV images from Google Maps print screens despite adding a series of control points measured with Google Earth Pro. However, Agisoft PhotosScan provided better results with images from Google Earth Pro than VisualSFM.

*Question: Is it the same for all the case studies? Any possible interpretation?*

*Answer: Sentence modified:* Different results are obtained as a function on the software used for SfM-MVS processing**. For all case studies**, VisualSFM gave the best results with print screens from GSV while Agisoft PhotoScan could not align any GSV images from Google Maps print screens despite adding a series of control points measured with Google Earth Pro. **Resolution of print screens images seem the be insufficient to be processed with Agisoft PhotoScan**. However, Agisoft PhotosScan provided better results with images from Google Earth Pro than VisualSFM.

*Page 5: Line 23*

The alignment of both point clouds was done on a stable part of the cliff, with a standard deviation of the error below 20 cm (Figure 3C).

*Question: What software has been used to htis end?*

*Answer: Paragraph modified:* "It was possible on "Basse Corniche" site to estimate the fallen volume by scaling and comparing the 2008 (**Figure 4A**) and 2010 (**Figure 4B**) point clouds. **The 2008 3D point cloud is composed of 150'000 points with an average density of 290 points per square meter and the 2014 3D point cloud is composed of 182'000 points with an average density of 640 points per square meter (Table 1**). VisualSFM software could align the images and make 3D models before and after the wall collapse. It was possible to roughly scale **and georeference** the scene with the road width **and few point coordinates** measured on Google Earth Pro and on the French geoportal (Géoportail, 2016). After aligning the two 3D point clouds, meshes were built to compute the collapsed volume. The **point to mesh** alignment **in CloudCompare software** of both point clouds was done on a **small** stable part of the cliff with a standard deviation of the error below 10 cm (Figure 4C) **and on the entire cliff beside the vegetation with a standard deviation of about 25 cm (Figure 4E).**"

*Page 6: Lines 7-9*

The number of 3D points on the landslide area varies from 9′500 to 25′000 points for a processing with VisualSFM, while 236′000 3D points were generated when using Agisoft PhotoScan.

*Question: What is the distance between points? Is the distance variating significantly as the distance from the camera increases?*

*Answer: in the Discussion (4.4), paragraph and figure (Figure 7) added: "***The point density was evaluated according to the distance between the image point of view and the subject and the image types and processing softwares. The obtained results and the derived trends indicate that the use of GSV images from Google Earth Pro with Visual SFM software increases of factor two the point density compared as the processing of GSV print screens with Visual SFM. The processing of GSV images from Google Earth Pro with Agisoft PhotoScan software increases of factor ten the point density compared as the processing of GSV print screens with Visual SFM (trend lines in Figure 7). Concerning the distance image point of view - area, the expected point density of the 3D point cloud from GSV print screens processed in Visual SFM software of a subject located few meters nears to the camera point of view ("Monaco" dots on Figure 7) is about 300 points/m2, about 50 points/m2 for an area located at about 100 m ("Arly" dots on Figure 7) and about 0.5 point/m2 for an area located at about 700 m ("Séchilienne" dots on Figure 7).***"*

*Figure 7 caption: "***Correlation between distance camera - case studies and the expected density of points from the three case studies. The red colour dots are results of the three case studies point clouds obtained from Google Street View (GSV) print screens (PS) in Google Maps (GM) processed with Visual SFM software. The red colour dash line represents their trend line based on the three case studies. The orange colour dot is the result of the Séchilienne point cloud obtained from GSV images saved in Google Earth Pro (GEP) processed with Visual SFM software. The orange colour dash line is its estimated trend line only based on the Séchilienne point cloud (point density multiplied by three compared to the red colour trend line). The green colour dots are results of the Séchilienne and Arly point clouds obtained from GSV images saved in (GEP) processed with Agisoft PhotoScan software. The green colour dash line is their estimated trend line based only on the Arly and Séchilienne point clouds (point density multiplied by eleven compared to the red colour trend line). By way of comparison, the blue colour dots represent the result of the Séchilienne and Arly point clouds obtained with GoPro action camera images taken on the field and processed with Agisoft PhotoScan software.***"*

*Page 6: Line 12*

(distance point to mesh in absolute values)

*Question: the absolute value would be 2.1 and not -2.1.*

*Answer: Paragraph modified. Those values were deleted.*

*Page 6: Line 23*

less accurate when using SfM-MVS processing

*Question: Please explain*

*Answer: I think that it is now understandable with the different added text in the manuscript that low resolution print screens with Visual SFM software provide less accurate results as images saved from Google Earth Pro an processed in Agisoft PhotoScan software.*

*Sentence modified: "Results were less accurate when using SfM-MVS processing with VisualSFM and lower resolution print screen images from Google Maps **probably due to the too low image resolution of those print screens**."*

*Page 7: Lines 16-17*

the GNSS integrated in the camera;

*Question: What about its scaling and orientation?*

*Answer: Sentence modified:" The 3D point cloud from the GoPro Hero5 Black images has been roughly georeferenced, **scaled and oriented** thanks to the GNSS **chip** integrated in the camera **and has been controlled and refined with point coordinate extracted from Google Maps and the French geoportal**."*

*Page 7: Line 20*

gives the least accurate results (Figure 5A).

*Question: please provide some quantitative information on the accuracy (level of error, point cloud density). How are the errors distributed all the point cloud, with respect to the distance from the photo camera?*

*Answer: it converges to the question "Page 6: Lines 7-9". In all paragraphs of section 4 "Results and discussion", there is now more information about the clouds comparison.*

*Page 9: Line 12-13*

According to the results, small-scale landslides and rockfalls (<1 m3) can be detected when the slope or the cliff is close to the road (0-10 m), as it was shown on site 1.

*Question: Are there areas where this small changes correspond to errors an although they have been detected, they are not realistic? Is their proportion important? Could you please comment on that?*

*Answer: Sentence further added: "**On such sites, small changes (<1 m3) can correspond to as well as realistic rockfalls as errors resulting of from processing like on the toe of the almost all Séchilienne landslide 3D point clouds (Figure 5 A2-H2).**"*

*Page 9: Lines 18-19*

This is attributable to the occurrence of slope movements generating material increase or decrease and thereby, increasing standard deviations of the error.

*Question: In the case of low density of the point cloud (of some meters of example), the roughness of the terrain in case study 2, due to the different sized of the deposited blocks plays an important role when aligning the point clouds and calculating the errors. How has this been taken into consideration, where the point cloud density is low?*

*Answer: Sentences added:* "**It can also be due to a bad 3D point cloud alignment. Indeed, the cloud alignments is not always easy on some point clouds because of low point density, because of voids in the point clouds (like in the landslide toe in Figure 5 F2) and because of the roughness of the terrain due to the different sized of the deposited blocks. In such difficult alignment cases, it was tried to align the point clouds on parts where the point cloud quality was the best to make an alignment and where the parts were stables.**"

*Page 17: Lines 1-5*

Figure 4: Results at site 2 "Séchilienne". Eight points clouds from different images sets taken at six different time with three different image sources and processed with two different programs. Figures A1-H1: Meshs resulting from the respective point clouds. Figures A2-H2: point clouds comparison with a 50 cm LiDAR DEM from 2010 (red colour points is material increase; blue colour points are material decrease from the 2010 LiDAR cloud). The information on the pictures source and date and on the program used is given in Table 1.

*Question: I think it would help to use the same colour scale for the easier comparison of the displacements at different point clouds.*

*Answer: All scales are now similar (-5 to +5 m).*

[revised manuscript text omitted]